# Leigh Syndrome in a Pedigree Harboring the m.1555A>G Mutation in the Mitochondrial 12S rRNA

**DOI:** 10.3390/genes11091007

**Published:** 2020-08-27

**Authors:** Mouna Habbane, Laura Llobet, M. Pilar Bayona-Bafaluy, José E. Bárcena, Leticia Ceberio, Covadonga Gómez-Díaz, Laura Gort, Rafael Artuch, Julio Montoya, Eduardo Ruiz-Pesini

**Affiliations:** 1Departamento de Bioquímica, Biología Molecular y Celular, Universidad de Zaragoza, 50013 Zaragoza, Spain; mounahabbane@gmail.com (M.H.); laurals@unizar.es (L.L.); pbayona@unizar.es (M.P.B.-B.); jmontoya@unizar.es (J.M.); 2Laboratoire Biologie et Santé, Faculté des Sciences Ben M’Sik, Université Hassan II, 20670 Casablanca, Morocco; 3Instituto de Investigación Sanitaria (IIS) de Aragón, 50009 Zaragoza, Spain; 4Centro de Investigaciones Biomédicas en Red de Enfermedades Raras (CIBERER), 28029 Madrid, Spain; lgort@clinic.cat (L.G.); rartuch@sjdhospitalbarcelona.org (R.A.); 5Servicio de Neurología, Hospital Universitario Cruces, 48903 Baracaldo, Vizcaya, Spain; joseeulalio.barcenallona@osakidetza.eus; 6Servicio de Medicina Interna, Hospital Universitario Cruces, 48903 Baracaldo, Vizcaya, Spain; leticia.ceberiohualde@osakidetza.eus; 7Servicio de Otorrinolaringología, Hospital Universitario Miguel Servet, 50009 Zaragoza, Spain; covagd@gmail.com; 8Errors Congènits del Metabolisme, Servicio de Bioquímica i Genètica Molecular, CDB, Hospital Clínic, IDIBAPS, 08036 Barcelona, Spain; 9Servicio de Bioquímica Clínica, Institut de Recerca Sant Joan de Déu, 08950 Barcelona, Spain; 10Fundación Araid, 50018 Zaragoza, Spain

**Keywords:** Leigh syndrome, hearing loss, mitochondrial DNA, ribosomal RNA, pathologic mutation, penetrance

## Abstract

Background: Leigh syndrome (LS) is a serious genetic disease that can be caused by mutations in dozens of different genes. Methods: Clinical study of a deafness pedigree in which some members developed LS. Cellular, biochemical and molecular genetic analyses of patients’ tissues and cybrid cell lines were performed. Results: mitochondrial DNA (mtDNA) m.1555A>G/*MT-RNR1* and m.9541T>C/*MT-CO3* mutations were found. The first one is a well-known pathologic mutation. However, the second one does not appear to contribute to the high hearing loss penetrance and LS phenotype observed in this family. Conclusion: The m.1555A>G pathological mutation, accompanied with an unknown nuclear DNA (nDNA) factor, could be the cause of the phenotypic manifestations in this pedigree.

## 1. Introduction

Leigh syndrome (LS) is a devastating neurodegenerative disease, typically manifesting in infancy or early childhood. Hallmarks of the disease are symmetrical lesions in the basal ganglia or brain stem on magnetic resonance imaging (MRI) and a clinical course with rapid deterioration of cognitive and motor functions [1].

This disorder can be caused by mutations in genes for mitochondrial DNA- (mtDNA) or nuclear DNA- (nDNA) encoded structural polypeptides of the oxidative phosphorylation (OXPHOS) complexes, nDNA-encoded OXPHOS assembly proteins and nDNA-encoded subunits unrelated to the OXPHOS complexes [2]. Mutations in mtDNA-encoded tRNA genes [3], or in genes for nDNA-encoded proteins related to mitochondrial translation, have been also frequently associated with LS [4]. However, to our knowledge, no LS patient has been previously associated with mutations in mtDNA-encoded rRNA genes. Confirmed rRNA pathologic mutations have been mainly involved in hearing loss [5,6].

In this manuscript, we report on two LS patients harboring the hearing loss-associated m.1555A>G mutation in the 12S rRNA (*MT-RNR1*) gene and perform a search for environmental and genetic factors that can explain the association of this mutation with the high prevalence of hearing loss in this pedigree, and with the LS phenotype in some of its members.

## 2. Materials and Methods

### 2.1. Ethical Issues

This study was approved by and carried out in accordance with the recommendations of the Institutional Review Board from the government of Aragón (CEICA CP-12/2014). Informed consent was obtained from all subjects, in accordance with the Declaration of Helsinki, and for publication of the case report.

### 2.2. Biochemical Studies

The mitochondrial respiratory chain (MRC) and citrate synthase (CS) enzyme-specific activities were measured in the muscle and fibroblasts with spectrophotometric procedures, as previously reported [7]. The expression of OXPHOS complexes was determined by blue native polyacrylamide gel electrophoresis analysis [8].

### 2.3. DNA Studies

DNA extraction, mtDNA sequencing, mtDNA copy number and m.1555A>G analyses were performed according to formerly established protocols [9,10,11]. To study the m.9541T>C mutation, the primers human mitochondrial L strand-9540-mispairing (hmtL-9540-mp) TCCAGCCTAGCCCCTACCCCCGAAT and hmtH9819 GCCAATAATGACGTGAAGTCC, and polymerase chain reaction conditions of 2 min at 94 °C (30 s, 94 °C/30 s, 56 °C/1 min 30 s, 72 °C) 30 cycles/5 min, 72 °C were used in amplification, and the enzyme PfeI (G/AWTC) at 37 °C was used in the enzymatic restriction.

### 2.4. Cybrids

Cybrid construction and analyses were performed as previously reported [12,13,14,15]. Changes in SDHA and p.MT-CO1 levels were normalized to a total cell number, measured according to manufacturer’s protocol using the Janus Green staining method (MitoBiogenesis^TM^ In-Cell ELISA Kit (Fluorescent)-Abcam, Cambridge, UK, ab140359).

### 2.5. Statistical Analysis

The statistical package StatView 6.0 was used to perform all the statistics. Data for mean and standard deviations were presented. The unpaired *t*-test was used to compare parameters. *p*-values lower than 0.05 were considered statistically significant.

## 3. Results

### 3.1. Clinical Cases

For familial antecedents, we reported on two cases that were born from young non-consanguineous parents. The older and younger siblings presented a similar severe phenotype, while the second sister was healthy. In the maternal family, there was a history of sensorineural deafness that led to a variable degree of hearing loss, affecting the mother, uncle, two aunts, the grandmother and a great-grandmother, as well as a great-grandaunt and her son. The grandmother and an aunt were exposed to aminoglycoside antibiotics, but there was no information in this respect about the other members of the pedigree. A maternal aunt presented a history of repeated miscarriages.

Patient V-2 of the pedigree started to develop motor clumsiness at four months old that was slowly progressive. Motor involvement was prominent while mild cognitive decline was observed. An MRI showed hyperintense signals, bilaterally and symmetrically affecting the caudate and putamen and, at the periaqueductal level, compatible with LS. Proton magnetic resonance with spectroscopy disclosed an increase in the lactate peak. X-ray exploration showed severe scoliosis at the lumbar level. Laboratory data displayed normal results, except for a mild decrement in free carnitine values.

In patient V-4, the first symptoms were observed in the second year of life, with gait delay associated with fatigability. The boy was wheelchair-bound and also presented pyramidal syndrome in his lower limbs, with gait disturbances, genu recurvatum and foot adduction. Cognitive evaluation displayed normal results, except for writing problems. No other signs or symptoms were observed. An MRI displayed bilateral hyperintense lesions, affecting the caudate and putamen, compatible with LS (Figure 1). The electroencephalogram and electromyogram displayed normal results. Laboratory data displayed normal results, except for a mild decrement in free carnitine values.

At present, the patients are 23 and 19 years old, and the clinical picture is slowly progressive.

### 3.2. Respiratory Complex Assays

In a muscle biopsy of the older sister (V-2), MRC complex IV (CIV)-specific activity was a third (20) of the lower limit of the control range (59) (Figure 2A), and the CIV/CS ratio was half (308) of the lower limit of the control range (590). The activities of the other respiratory complexes were not significantly diminished. The determination of the quantity of OXPHOS complexes showed a moderate reduction in CIII (81%) and CV (70%), in respect to a control muscle. Moreover, there were CV sub-complexes (Figure 2B). However, the CIV levels were 26% of those from the control sample.

The fibroblast MRC CIV-specific activity and CIV/CS ratio from the younger brother (V-4) were close to the lower limit of the reference range.

### 3.3. Molecular-Genetic Analysis

The pedigree analysis (Figure 3A), and the reduced levels in CIV activity and quantity, led us to sequence the whole mtDNA from the muscle of V-2, obtaining the following homoplasmic genetic variants: m.146T>C, m.263A>G, m.309-310insC, m.315-316insC, m.750A>G, m.1438A>G, m.1555A>G, m.3010G>A, m.4769A>G, m.8860A>G, m.9356C>T, m.9541T>C, m.15326A>G, m.16278C>T and m.16519T>C (GenBank, MT734693). Thirteen of these variants define mtDNA haplogroup H1r1 [16]. This sequence included the pathologic mutation m.1555A>G in the *MT-RNR1* gene (Figure 3B) that had been previously associated with antibiotic-induced and non-syndromic deafness [5]. This mutation was homoplasmic in all members of the pedigree (Figure 3C) and can explain the hearing loss phenotype in maternally related individuals. However, it did not explain the high penetrance of maternally inherited deafness (11 out of 14 maternal relatives developed hearing loss) nor LS.

As we had no access to the cochlear hair cells, we determined the mtDNA copy number by blood. The blood mtDNA copy number of V-2 was the lowest of the pedigree. The unaffected V-3 and V-5 individuals showed high mtDNA copy numbers, but the LS patient, V-4, also presented high mtDNA levels (Appendix A). Therefore, the mtDNA copy number did not explain this LS phenotype. We must consider that age is important in studies of mtDNA levels, but this factor is not easy to control in a pedigree.

The mtDNA sequence also harbored a private mutation, m.9541T>C, in the *MT-CO3* gene (Figure 3D). The mutation has been previously found thrice in three different haplogroups out of 51,192 mtDNA sequences [17]. To rule out this mutation as a geographically restricted polymorphism, we also studied 37 Spanish population samples from the same mtDNA haplogroup (H), and it was not found. The mutation replaces a leucine at amino acid position 112 of p.MT-CO3 with a serine. According to the pathogenicity predictors MutPred, PolyPhen-2 and Mitoclass.1, this amino acid substitution is considered pathogenic [18,19]. Moreover, the L112 amino acid is conserved in 78.6% of 4826 p.MT-CO3 reference sequences, including animal, plant, fungi and protist species [19]. This conservation is higher than the mean conservation of the whole p.MT-CO3 protein (72.8%).

We analyzed the mutation in maternally related individuals. The blood from the grandmother (III-2) and from all the maternally related individuals of generations IV and V was homoplasmic for this mutation (Figure 3E). Muscle from IV-2 and fibroblasts from IV-4 were also homoplasmic for this change.

### 3.4. Cybrids

To obtain evidence on the functional effects of m.9541T>C, we generated a transmitochondrial cell line, cytoplasmic hybrid—or a cybrid with the osteosarcoma 143B nuclear genetic background—and the mtDNA of patient V-2 (P), harboring the m.1555A>G and m.9541T>C mutations, as well as a positive control (PC) harboring the m.1555A>G transition and a negative control (NC) without any of these mutations.

ATP levels, oxygen consumption (endogenous, leaking or uncoupling), CIV and CS-specific activities, CV/CS, in-gel CIV, in-gel complex I + II (CI+II), CIV quantity, p.MT-CO1/JG and SDHA/JG were not significantly decreased in P when compared to PC (Appendix A). 

## 4. Discussion

The two more interesting facts about this m.1555A>G pedigree are the very high penetrance of sensorineural hearing loss (SNHL) and the LS phenotype in two patients.

It has been shown that mtDNA genetic variability can modulate the phenotype expression of m.1555A>G mutations. Thus, a large Han Chinese pedigree carrying both m.1555A>G and m.4317A>G mutations exhibited much higher SNHL penetrance than those carrying only the m.1555A>G mutation. In lymphoblastoid cell lines, the combination of both mutations provoked a more severe mitochondrial dysfunction than that caused by m.1555A>G alone [20].

Mitochondrial abnormalities associated with m.1555A>G mutations may be expressed in tissues other than those of the auditory system. Thus, muscle biopsies of three SNHL patients showed disorganized intermyofibrillar networks with atypical morphology, a few typical ragged red fibers and decreased CIV activity [21,22]. This homoplasmic mutation has also been associated with SNHL and myelocystocele–cloacal exstrophy, pigmentary disturbances and spinal anomalies. The syndromic features of this family appeared to co-segregate with the m.1555A>G mutation [23]. This homoplasmic m.1555A>G mutation has furthermore been associated with SNHL and hypertension [24], SNHL and diabetes [25] and SNHL and cardiomyopathy [26,27]. A woman suffering from maternally inherited cardiomyopathy without SNHL in her pedigree showed slightly decreased activity of the MRC complexes in muscle and skin fibroblast homogenates when values were normalized to the activity of CS and harbored the m.1555A>G mutation in heteroplasmy. The percentage of heteroplasmy was higher in abnormal muscle fibers than normal ones [28]. The mtDNA genetic variability associated with the m.1555A>G mutation can be responsible for additional phenotypes associated with SNHL. Thus, a patient harboring the m.1555A>G and m.4309G>A mutations showed SNHL, progressive external ophthalmoplegia and exercise intolerance [29]. 

Using cybrid cell lines, we have not found differences between the double mutation (m.1555A>G and m.9541T>C) and the single one (m.1555A>G). Therefore, despite there being some evidence for m.9541T>C mutation being pathogenic, cybrid results seem to discard this possibility and, therefore, this mutation does not appear to be responsible for increased SNHL penetrance, although we can never rule out that its presence in tissues could have an effect. A possible way to confirm a tissue-specific effect would be using hair cell-like cells differentiated from the induced pluripotent stem cells of the patient [30]. In relation to LS in this pedigree, environmental or nDNA factors must be involved in its appearance. Unfortunately, we have no detailed information on aminoglycoside exposure of the LS patients’ mother (IV-4). However, increasing evidence indicates that long-lasting effects on the vestibular organs and kidneys can be induced upon long prenatal exposure to high concentrations of aminoglycosides [31,32]. Perhaps these drugs could also have negative effects on the brain development of embryos carrying the m.1555A>G mutation. Another possibility to explain the LS phenotype in this family would involve nDNA factors. We previously noted that amino acid substitutions in the nDNA-encoded mitochondrial ribosomal protein S12, or other ribosomal proteins that interact with the mtDNA-encoded 12S rRNA, were potential candidates to explain the altered penetrance of rRNA homoplasmic mutations [33]. Conceivably, aberrant interactions between nDNA- and mtDNA-encoded factors might be common causes of mitochondrial diseases [34].

## Figures and Tables

**Figure 1 genes-11-01007-f001:**
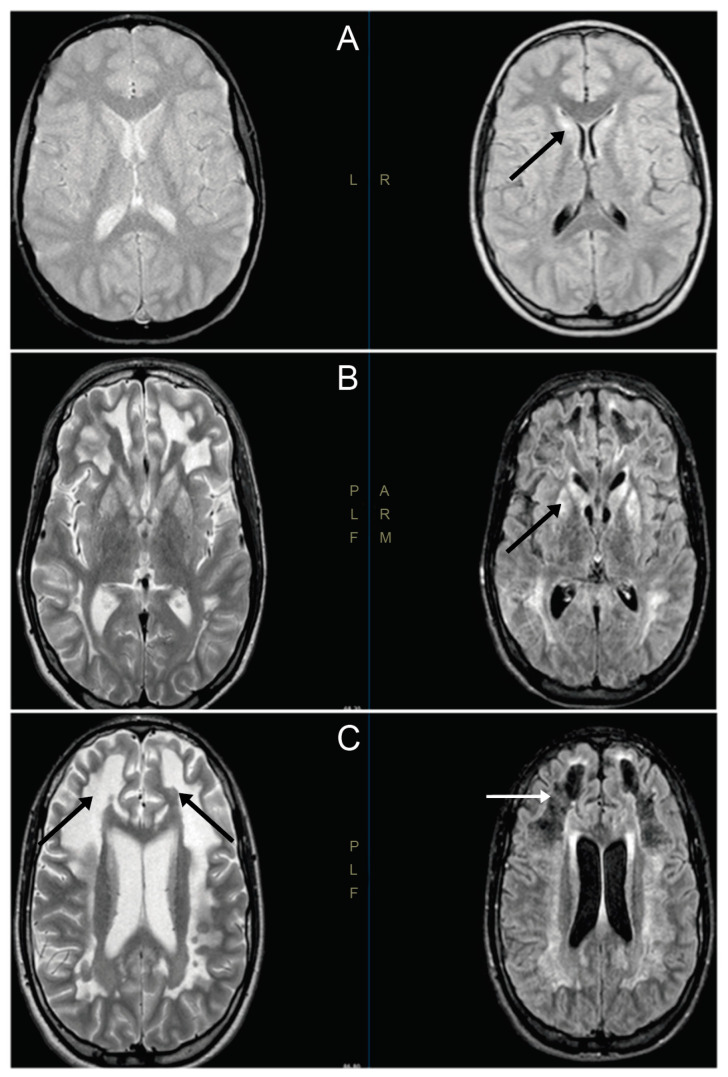
Magnetic resonance images of V-4′s brain. (**A**) Cranial magnetic resonance images (MRIs) at 9 years old. Hyperintense lesions are observed symmetrically, affecting both heads of the caudate and putamen. The putamen-focused spectroscopic study showed a peak of high lactate, with a decrease in N-acetylaspartate and creatine (black arrow). (**B**) Cranial MRI at 15 years old, showing an increase in lesions in the basal ganglia (black arrow). (**C**) Cranial MRI at 15 years old, showing great involvement of white matter demyelination in both hemispheres (black arrows) and in the area of the frontal necrotic lesions (white arrow).

**Figure 2 genes-11-01007-f002:**
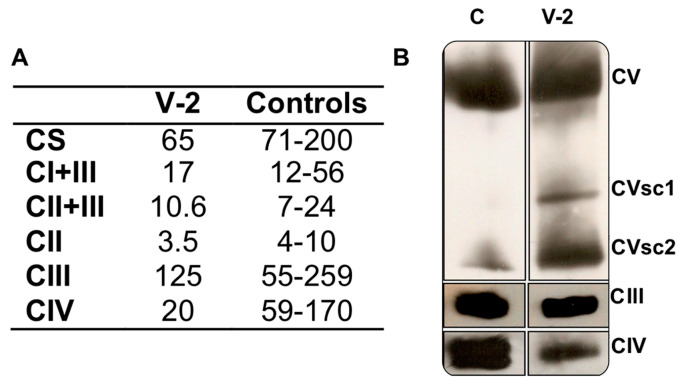
Muscle biochemistry. (**A**) Respiratory complex I + III (CI + III), II + III (CII + III), II (CII), III (CIII), IV (CIV) and citrate synthase- (CS) specific activities (nmol/min x mg protein) of the Leigh syndrome (LS) patient (V-2) and controls. (**B**) Respiratory complex levels of V-2 and a control (**C**). CVsc1 and 2 are code for CV sub-complexes 1 and 2, respectively.

**Figure 3 genes-11-01007-f003:**
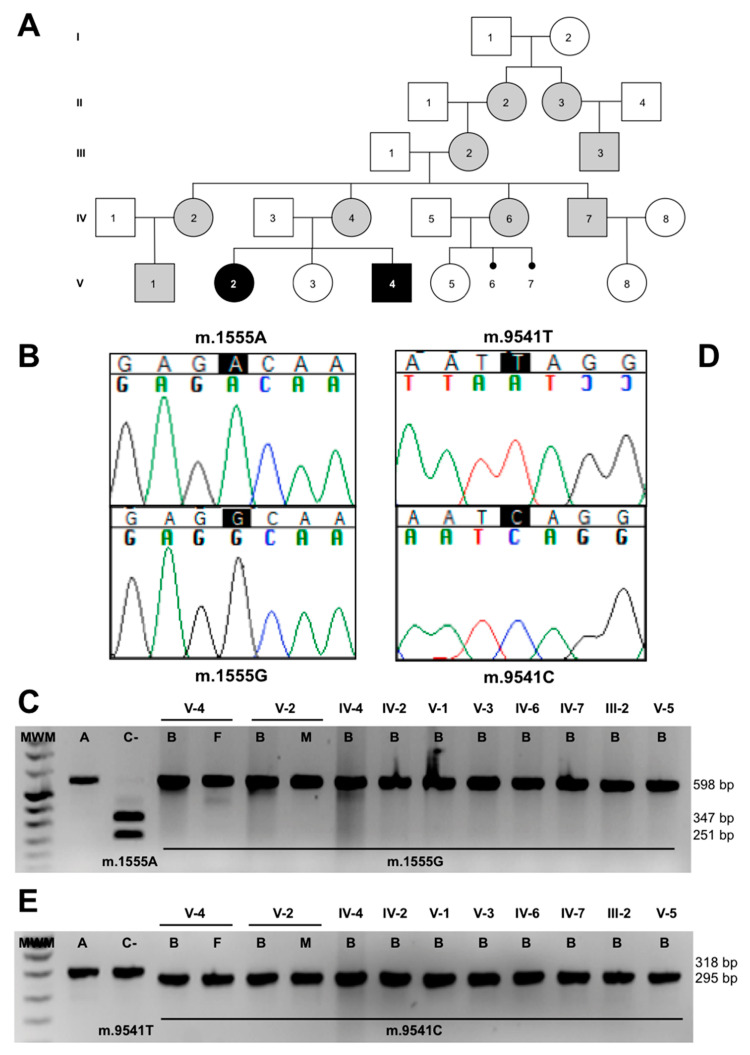
m.1555A>G and m.9541T>C mutations. (**A**) Pedigree. Gray and black colors are code for hearing loss and Leigh syndrome patients, respectively. (**B**) Electropherogram of a segment of the patient’s mtDNA sequence, showing the m.1555A>G transition. (**C**) Restriction fragment length polymorphism (RFLP) gel, showing the mutation load in the pedigree members. (**D**) Electropherogram of a segment of the patient’s mtDNA sequence, showing the m.9541T>C transition. (**E**) RFLP gel, showing the mutation load in the pedigree members. A = amplicon, C- = negative control, B = blood, F = fibroblasts and M = muscle.

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
