# Peer review of "Leigh Syndrome in a Pedigree Harboring the m.1555A>G Mutation in the Mitochondrial 12S rRNA"

_genes, 2020, doi:10.3390/genes11091007_

Round 1
Reviewer 1 Report
This case report by Habbane and colleagues describe a rather large pedigree carrying the common 1555A>G mutation in the MT-RNR1 gene, which is usually associated with antibiotic-induced deafness. However, several mambers of this family were affected by sensorineural hearing loss. In addition, two siblings from this non-consaguineous family were affected by Leigh syndrome. One of the two had complex IV deficiency in the skeletal muscle, whilst the other had COX activity at the lower limit of the controls.
The analysis of mtDNA found a number of variants, including the homoplasmic 1555A>G mutation and a 9541T>C mutation in MT-CO3 gene, encoding complex IV core subunit 3, which is predicted to be pathogenic. Also this mutation was, however, homoplasmic in all the family members.
Analysis of cybrid cells did not reveal any difference between cells carrying only the 1555A>G and those carrying both 1555A>G and 9541T>C mutations.
This isan interesting case report, but I have a couple of concerns. First, the authors report that a number of other variants were detected but the authors do not comment on them.
Second, the absence of a phenotype in cybrids carrying the double mutation does not exclude a possible tissue-specific effect. Although the authors comment on this in the discussion, I think that they should at least discuss how this issue could be tackled in future studies.
Fianlly, I strongly reccommend the authors to carefully check the English as tehre are many mistakes that should be corrected.
Author Response
Reviewer 1.
This case report by Habbane and colleagues describe a rather large pedigree carrying the common 1555A>G mutation in the MT-RNR1 gene, which is usually associated with antibiotic-induced deafness. However, several mambers of this family were affected by sensorineural hearing loss. In addition, two siblings from this non-consaguineous family were affected by Leigh syndrome. One of the two had complex IV deficiency in the skeletal muscle, whilst the other had COX activity at the lower limit of the controls.
The analysis of mtDNA found a number of variants, including the homoplasmic 1555A>G mutation and a 9541T>C mutation in MT-CO3 gene, encoding complex IV core subunit 3, which is predicted to be pathogenic. Also this mutation was, however, homoplasmic in all the family members.
Analysis of cybrid cells did not reveal any difference between cells carrying only the 1555A>G and those carrying both 1555A>G and 9541T>C mutations.
This isan interesting case report, but I have a couple of concerns. First, the authors report that a number of other variants were detected but the authors do not comment on them.
Fifteen single nucleotide polymorphisms (SNPs) were different in the patient’s sequence versus the revised Cambridge Reference Sequence (rCRS1). Thirteen of them defined mtDNA haplogroup H1r12. Only m.1555A>G and m.9541T>C were private mutations and were analyzed and commented in the manuscript. This is now mentioned in the text (line 203). We have changed “This genotype belongs to mtDNA haplogroup H1r1” by “Thirteen of these variants define mtDNA haplogroup H1r1”. We have also included the GenBank code (MT734693) in the line 203.
1 Andrews et al, 1999. Reanalysis and revision of the Cambridge reference sequence for human mitochondrial DNA. Nat Genet 23, 147. PMID: 10508508.
2 van Oven et al, 2009. Updated comprehensive phylogenetic tree of global human mitochondrial DNA variation. Hum Mutat 30, E386-94. PMID: 18853457.
Second, the absence of a phenotype in cybrids carrying the double mutation does not exclude a possible tissue-specific effect. Although the authors comment on this in the discussion, I think that they should at least discuss how this issue could be tackled in future studies.
We totally agree with the referee and, as he/she observes, we already mentioned it in the “Discussion” section. This is a very difficult issue. Currently, it is not possible to make site directed mutagenesis in the mtDNA. Therefore, the only ways to confirm a tissue-specific effect of the double mutation is having this tissue (cochlear hair cells) from the patient or using hair cell-like cells differentiated from induced pluripotent stem cells (iPSCs) of the patient. Both possibilities are very complicated because require the patients’ active participation, who are often no longer interested in investigating their diseases after years of wasted efforts, and an efficient protocol of differentiation to this particular cell type. In any case, we have now mentioned that iPSCs can help in the future to unravel this question (“A possible way to confirm a tissue-specific effect would be using hair cell-like cells differentiated from induced pluripotent stem cells of the patient”) and included the next reference1 in lines 334-336.
1 Czajkowski et al, 2019. Pluripotent stem cell-derived cochlear cells: a challenge in constant progress. Cell Mol Life Sci 76, 627-35. PMID: 30341460.
Fianlly, I strongly reccommend the authors to carefully check the English as tehre are many mistakes that should be corrected.
We have now carefully checked the English to correct previous mistakes.
Reviewer 2 Report
The manuscript by Habbane and colleagues describe a family with the mitochondrial DNA (mtDNA) m.1555A>G/MT-RNR1 and a second m.9541T>C/MT-CO3 mutations. The m.1555A>G is known to cause non-syndromic deafness. They concluded that the m.9541T>C mutation does not appear to contribute to the high hearing loss penetrance and LS phenotype observed in this family. They conclude that the nuclear background explains the phenotypic differences.
It is an interesting study that widens the phenotypic variability of the m.1555A>G mutation. I have relatively minor comments.
- For the analysis of mtDNA levels, age is critical. That makes the intrafamilial comparisons difficult. Were controls age matched?
- Why there are no westerns for complex I subunits?
- “CV/CS, in gel CIV, in gel complex I + II (CI+II), CIV quantity, 276 p.MT-CO1/JG or SDHA/JG were not importantly decreased in P when compared to 277 PC (Figure S2).” Importantly is not a scientific term when describing results
- “Using cybrid cell lines, we have not found differences between the double mutant (m.1555A>G and m.9541T>C) and the single one (m.1555A>G). Therefore, despite that there was some evidences for m.9541T>C mutation being pathogenic, cybrid results seems to discard this possibility and, therefore, this mutation does not appear to be responsible for the increased SNHL penetrance.” It is hard to use the cybrids as evidence for the synergistic role of these variations, as they did not show any phenotype to start with.
Author Response
The manuscript by Habbane and colleagues describe a family with the mitochondrial DNA (mtDNA) m.1555A>G/MT-RNR1 and a second m.9541T>C/MT-CO3 mutations. The m.1555A>G is known to cause non-syndromic deafness. They concluded that the m.9541T>C mutation does not appear to contribute to the high hearing loss penetrance and LS phenotype observed in this family. They conclude that the nuclear background explains the phenotypic differences.
It is an interesting study that widens the phenotypic variability of the m.1555A>G mutation. I have relatively minor comments.
For the analysis of mtDNA levels, age is critical. That makes the intrafamilial comparisons difficult. Were controls age matched?
We totally agree with the referee. It is not an easy thing to match ages in pedigree studies and we could not make it. We have now included this consideration in the “3.3 Molecular-genetic analysis” section, lines 265-267 (“although we must consider that age is important in studies of mtDNA levels but this factor is not easy to control in a pedigree”).
Why there are no westerns for complex I subunits?
Now it is possible to determine all the OXPHOS complexes in the same Western blot analysis. When these analyses were performed, several gels had to be run to determine by Western blot all the OXPHOS complexes. The patient’s sample was very tiny and run out.
“CV/CS, in gel CIV, in gel complex I + II (CI+II), CIV quantity, 276 p.MT-CO1/JG or SDHA/JG were not importantly decreased in P when compared to 277 PC (Figure S2).” Importantly is not a scientific term when describing results
We have now changed “importantly” by “significantly” (line 295).
“Using cybrid cell lines, we have not found differences between the double mutant (m.1555A>G and m.9541T>C) and the single one (m.1555A>G). Therefore, despite that there was some evidences for m.9541T>C mutation being pathogenic, cybrid results seems to discard this possibility and, therefore, this mutation does not appear to be responsible for the increased SNHL penetrance.” It is hard to use the cybrids as evidence for the synergistic role of these variations, as they did not show any phenotype to start with.
Bioenergetics differences between the mutant (PC) and double mutant (P) cybrids might provide evidences of the synergistic role of these variations, in particular, a decrease in OXPHOS parameters in P versus PC. In fact, functional studies using cybrids have been considered high quality evidence in support of pathogenicity1.
1 Mitchell et al, 2006. Sequence variation in mitochondrial complex I genes: mutation or polymorphism? J Med Genet 43, 175-9. PMID: 15972314.
Reviewer 3 Report
The manuscript by Mouna Habbane and coworkers presents and describes a patient with Leigh syndrome, caused by an unusual combination of mitochondrial DNA (mtDNA) mutations. For example, the A1555G has been most commonly associated with hearing loss, which is much more benign than the devastating multisystemic Leigh syndrome.
The manuscript is generally well written and data well presented. I have only few minor points.
L66: Typo: “none LS patient”
L164: “third and a half”. Please give precise relative numbers. Do the controls really vary as much as in Figure 1A. What is considered “normal” range here? I see that no statistical analysis can be made with the samples, but please provide median numbers for the controls together with the range.
L192: Whole mtDNA sequencing is not explained in the methods?
L192: Check word choice: “drove us to”
L193: Were the other variants than A1555G also homoplasmic?
L255: use “replaces” instead of “changes”
L269: “built” -> created/generated.
LS should be progressive, yet the patients are described to have a stable clinical picture (L130-1). I think this needs to be elaborated/discussed further.
L291: “moth-eaten” sounds very wrong when talking about patients. Please modify. The pathology suggests that this is a genuine mitochondrial syndrome, however the fact that cybrid cells do not have a phenotype and the muscle biochemistry findings are also fairly small, suggests that the etiology must relate to a nuclear gene defect? Please make this more clear if you agree, otherwise explain the findings better. Is there any indication of aminoglycoside exposure during pregnancy, which might have caused (via A1555G) the defect during the brain development? This could explain why the syndrome is not progressive.
Author Response
The manuscript by Mouna Habbane and coworkers presents and describes a patient with Leigh syndrome, caused by an unusual combination of mitochondrial DNA (mtDNA) mutations. For example, the A1555G has been most commonly associated with hearing loss, which is much more benign than the devastating multisystemic Leigh syndrome.
The manuscript is generally well written and data well presented. I have only few minor points.
L66: Typo: “none LS patient”
The word “none” has been changed to “no” (line 69).
L164: “third and a half”. Please give precise relative numbers. Do the controls really vary as much as in Figure 1A. What is considered “normal” range here? I see that no statistical analysis can be made with the samples, but please provide median numbers for the controls together with the range.
As it can be observed in the table of the Figure 2A, the CIV specific activity was a third (20 nmol / min x mg protein) of the lower limit of the control range (59 nmol/min x mg protein). Real values for CIV/CS were 308 and 590 for the patient and the lower limit of the control range, respectively. These values are now included in the text (lines 169 - 171). We suppose that the referee is referring to Figure 2A when he/she mentions the variation in control individuals. Yes, these values determined the normal range for muscle CIV specific activity in control individuals obtained in our laboratory. Normal range provides the specific activities in muscle from healthy individuals. These clinical ranges were obtained many years ago and we do not have the median numbers.
L192: Whole mtDNA sequencing is not explained in the methods?
As referenced in the “Material and Methods” section (line 89), mtDNA sequencing was explained in Supplementary Table 2 from one of the cited references1.
1 Emperador et al, 2018. The decrease in mitochondrial DNA mutation load parallels visual recovery in a Leber hereditary optic neuropathy patient. Front Neurosci 12, 61. PMID: 294793.
L192: Check word choice: “drove us to”
We have changed “drove us to” by “led us to” (line 199).
L193: Were the other variants than A1555G also homoplasmic?
All mtDNA variants were apparently homoplasmic, according to the sequencing electropherograms (line 200). All, except m.1555A>G and m.9541T>C polymorphisms define mtDNA haplogroup H1r1 and should be homoplasmic1. The private mutations m.1555A>G and m.9541T>C were checked, as it was described in the manuscript, and they were also homoplasmic.
1 van Oven et al, 2009. Updated comprehensive phylogenetic tree of global human mitochondrial DNA variation. Hum Mutat 30, E386-94. PMID: 18853457.
L255: use “replaces” instead of “changes”
We have substituted “changes” by “replaces” (line 274).
L269: “built” -> created/generated.
We have substituted “built” by “generated” (line 288).
LS should be progressive, yet the patients are described to have a stable clinical picture (L130-1). I think this needs to be elaborated/discussed further.
We agree with the referee comment, since we did not properly explain the patient evolution. We meant that the patients reached adulthood and are still alive but the clinical picture was slowly progressive (no stable). In fact, in the Figure 1 legend, we already stated that the MRI showed a progressive evolution at 15 years old (panels B and C) when compared with 9 years old (panel A). Thus, we have changed the sentence “At present, the patients are 23 and 19 years old and the clinical picture remain stable” by “At present, the patients are 23 and 19 years old and the clinical picture is slowly progressive” (lines 134-135).
L291: “moth-eaten” sounds very wrong when talking about patients. Please modify.
The expression “moth-eaten” was literally taken from the original article1. We have also found this expression in other scientific articles2,3. In any case, we have changed “a moth-eaten appearance” by “disorganized intermyofibrillar networks” (line 313).
1 Yamasoba et al, 2002. Atypical muscle pathology and a survey of cis-mutation in deaf patients harboring a 1555 A-to-G point mutation in the mitochondrial ribosomal RNA gene. Neuromuscul Disord 12, 506-12. PMID: 12031626.
2 Goto et al, 1990. Chronic progressive external ophthalmoplegia: a correlative study of mitochondrial DNA deletions and their phenotypic expression in muscle biopsies. J Neurol Sci 100, 63-9. PMID: 1965208.
3 Arenas et al, 1998. Complex I defect in muscle from patients with Huntington’s disease. Ann Neurol 43, 397-400. PMID: 9506560.
The pathology suggests that this is a genuine mitochondrial syndrome, however the fact that cybrid cells do not have a phenotype and the muscle biochemistry findings are also fairly small, suggests that the etiology must relate to a nuclear gene defect? Please make this more clear if you agree, otherwise explain the findings better.
We agree with this referee and we already commented it in the last paragraph of the “Discussion” section, “nDNA factors must be involved in its appearance”. Perhaps a potential nDNA/mtDNA incompatibility might be the cause of the LS phenotype in this pedigree1. We have now tried to make this point clearer (lines 343-351).
1 Potluri et al, 2009. A novel NDUFA1 mutation leads to a progressive mitochondrial complex I-specific neurodegenerative disease. Mol Genet Metab 96, 189-95. PMID: 19185523.
Is there any indication of aminoglycoside exposure during pregnancy, which might have caused (via A1555G) the defect during the brain development? This could explain why the syndrome is not progressive.
Unfortunately we have not detailed information on aminoglycoside exposure in this pedigree. However, this is a very interesting possibility and we have now included it in the text (lines 337-343). Increasing evidence points that long-lasting effects on vestibular organ and kidney can be induced upon long prenatal exposure to high concentrations of aminoglycosides1,2. Perhaps these drugs could also have negative effects on the brain development of embryos carrying the m.1555A>G mutation.
1 Deguine et al, 1996. Prenatal lesioning of vestibular organ by aminoglycosides. Neuroreport 7, 2435-8. PMID: 8981398.
2 Mantovani et al, 2001. Delayed developmental effects following prenatal exposure to drugs. Curr Pharm Des 7, 859-80. PMID: 11375782.
Round 2
Reviewer 1 Report
The authors addressed my concerns and I don't have additional issues to raise.